# The Role of Preservation Solutions upon Saphenous Vein Endothelial Integrity and Function: Systematic Review and UK Practice Survey

**DOI:** 10.3390/cells12050815

**Published:** 2023-03-06

**Authors:** Georgia R. Layton, Shameem S. Ladak, Riccardo Abbasciano, Liam W. McQueen, Sarah J. George, Gavin J. Murphy, Mustafa Zakkar

**Affiliations:** 1Department of Cardiovascular Sciences, University of Leicester, Leicester LE1 7RH, UK; 2Department of Cardiac Surgery, Imperial College London, London SW7 2BX, UK; 3Department of Translational Health Sciences, Bristol Medical School, University of Bristol, Bristol BS2 1UDD, UK

**Keywords:** vein graft disease, coronary artery bypass grafting, venous endothelium

## Abstract

The long saphenous vein is the most used conduit in cardiac surgery, but its long-term patency is limited by vein graft disease (VGD). Endothelial dysfunction is a key driver of VGD; its aetiology is multi-factorial. However emerging evidence identifies vein conduit harvest technique and preservation fluids as causal in their onset and propagation. This study aims to comprehensively review published data on the relationship between preservation solutions, endothelial cell integrity and function, and VGD in human saphenous veins harvested for CABG. The review was registered with PROSPERO (CRD42022358828). Electronic searches of Cochrane Central Register of Controlled Trials, MEDLINE, and EMBASE databases were undertaken from inception until August 2022. Papers were evaluated in line with registered inclusion and exclusion criteria. Searches identified 13 prospective, controlled studies for inclusion in the analysis. All studies used saline as a control solution. Intervention solutions included heparinised whole blood and saline, DuraGraft, TiProtec, EuroCollins, University of Wisconsin (UoW), buffered, cardioplegic and Pyruvate solutions. Most studies demonstrated that normal saline appears to have negative effects on venous endothelium and the most effective preservation solutions identified in this review were TiProtec and DuraGraft. The most used preservation solutions in the UK are heparinised saline or autologous whole blood. There is substantial heterogeneity both in practice and reporting of trials evaluating vein graft preservation solutions, and the quality of existing evidence is low. There is an unmet need for high quality trials evaluating the potential for these interventions to improve long-term patency in venous bypass grafts.

## 1. Introduction

The long saphenous vein (LSV) is the most used conduit in cardiac surgery; however, its use is complicated by an increased risk of late stenosis or occlusion due to the development of intimal hyperplasia (IH) and accelerated atherosclerosis; a pathology comprehensively described as vein graft disease (VGD) [1,2]. Endothelial dysfunction is a key driver of VGD due to propagation of an immune response, inflammatory activation, and cellular differentiation processes. Whilst its causation is multi-factorial, conduit harvest technique and preservation fluids have been identified as critical culprits in its onset and propagation [3]. Once surgically harvested, the vein is submerged in a solution of the surgeon’s choice until it is ready to be implanted for bypass grafting. The solutions used vary by surgeon. To date, no solution has been widely accepted as superior. Furthermore, the specific preferences of surgeons across the UK are currently unknown and there is no robust evidence or guidance as to which solutions should be preferentially used [4]. This review aims to assess the relationship between preservation solutions and endothelial cell (EC) integrity and function in human saphenous veins being used for coronary artery bypass grafting (CABG). Moreover, we conducted a national survey of all cardiothoracic surgery units in the UK to establish the current individual use of preservation solutions for intra-operative preservation of venous conduits.

## 2. Materials and Methods

This systematic review was performed following guidance from the Preferred Reporting Items for Systematic Reviews and Meta Analyses (PRISMA) statement standard [5]. A study protocol was designed which conformed to the PRISMA protocol standard [6] and was registered at the International Prospective Register of Systematic Reviews (PROSPERO ID CRD42022358828) [7].

### 2.1. Study Eligibility

The inclusion criteria were:Any studies utilising preservation solutions during or following the process of saphenous vein harvest;Analysing the effect of these solutions on saphenous vein endothelial cell integrity, function, or both, versus a control. Accepted control solutions were saline with or without heparin;Human subjects undergoing coronary artery bypass grafting; andAll study models (in vivo, in vitro, and ex vivo).

Exclusion criteria included:5.No implementation of preservation solutions;6.No provision of a control group who received standard of care, as defined by study authors;7.Non-human subjects;8.Analysing arterial endothelial cell integrity or function;9.Systematic review or meta-analyses not reporting original data;10.Conference and meeting abstracts, case reports and literature reviews;11.Studies not published in English.

### 2.2. Search Strategy

Electronic searches were conducted using the Cochrane Central Register of Controlled Trials, MEDLINE, and EMBASE without date or language restriction from inception until August 2022 (see Appendix A).

The search strategy employed to determine studies of relevance utilised combinations of keywords such as “saphenous vein”, “coronary artery bypass grafting”, and specific named preservation solutions utilised in practice like “University of Wisconsin”. A full description of the search strategy is listed in the Appendix A. In addition, the reference lists of all retrieved articles were searched for further relevant studies not previously identified.

Only papers that were published available in English were considered for subsequent analysis. References from selected papers were searched for relevant articles to ensure the literature search was thorough. Reviewer R.A. performed the database searches. Search results were imported into the Rayyan QCRI web app [8], and duplicates were identified and removed.

To select relevant papers identified by the electronic search, papers were assessed initially by their title, then by analysis of their abstracts. Reviewers G.R.L, S.L and M.Z performed this independently. Conflicts were resolved by consensus discussion between all three reviewers. Studies not excluded after this stage were then examined in full to assess their relevance. Authors G.R.L, S.L and M.Z then validated the final selected papers and conflicts were resolved by consensus discussion.

### 2.3. Data Extraction

A standardised form was developed to extract data from the included studies for assessment of study quality and evidence synthesis. This form was tabulated using Microsoft Excel 2016 (Microsoft, Redmond, WA, USA).

Data extraction first considered data from figures, tables, and graphs (using digital image analyser software Webplot Digitizer [9] as necessary), followed by data extraction from the main text. Data was collected as standardised mean differences for continuous outcomes and median and quartile ranges for categorical outcomes. Data extracted to the standardised form were categorised under the following headings: title, author, year, journal, study design, methodology including surgical harvest technique, population demographics (if reported), methodology, clinical outcome measures (if reported), endothelial integrity outcome measures and endothelial function outcome measures. Reviewer G.R.L performed data extraction, authors G.R.L, S.L and M.Z validated the findings, and any discrepancies were resolved by discussion.

### 2.4. Study Outcomes

The primary outcome measure was endothelial integrity or function. Secondary outcome measures included clinical outcomes after CABG that are known to be associated with the development of vein graft disease including: recurrence of symptoms, need for repeat revascularisation, mortality, and major adverse cardiovascular events (MACE). Bias and Quality Assessment Quality of included studies was assessed using the ROBINS-I tool [10]. Following data extraction, reviewer G.R.L performed quality and risk of bias assessment on all studies. Any discrepancies were resolved by discussion between all authors.

### 2.5. Data Synthesis

A narrative synthesis of all included studies was performed with all relevant data tabulated where appropriate. For all outcomes, data were extracted in text format or as mean ± standard deviation for numerical values. Where applicable, continuous variables were summarised with standardised mean difference. Given the anticipated diversity of outcome measures, limited scope for statistical analysis was expected and as such, meta-analysis was not undertaken.

### 2.6. Survey of National Practice

A cross-sectional, electronic survey was distributed by the national specialty society SCTS, between 16th August–16th September 2022, as convenience sampling of all adult cardiac surgery professionals in the UK. A combination of open and closed questioning was used to explore current practices and beliefs regarding vein graft integrity in each unit offering adult cardiac surgery in the UK (Appendix B).

## 3. Results

A total of 229 articles were identified by searches. Sixty-three were removed following the automated removal of duplicates by Rayaan. 166 were screened against the inclusion/exclusion criteria. Of these papers, 13 were eligible for final synthesis (Figure 1). A summary of the characteristics of all included studies are reported in Table 1. The included studies span more than four decades (1980–2022). All studies prospectively assessed human saphenous vein following harvesting for CABG. There was large heterogeneity of sample size, ranging from 5 to 125 participants. Most studies harvested the vein in an open fashion, two studies also used endoscopic approaches, and four did not explicitly describe the specific harvest method. All studies did however report standardisation of harvest technique across study samples.

There were nine different preservation solutions used including heparinised whole blood, DuraGraft, TiProtec, EuroCollins, University of Wisconsin, Pyruvate, buffered cell culture solution with albumin (M199 with HEPES), cardioplegic and papaverine solutions. The compositions of these solutions are outlined in Table 2. All studies compared solutions against a saline control. In addition, six studies also assessed the impact of concurrent intervention with varied time of exposure to solution (*n* = 4), distention pressure (*n* = 4) or anatomical position within the vein from which the study sample was harvested (*n* = 1). All studies addressed endothelial integrity, with most utilising differing methods of outcome assessment. Three studies utilised a shared outcome assessment methodology to analyse endothelial-dependent vasoreactivity [16,22,23].

The included studies were published in eleven journals with impact factors ranging from 0.7 to 5.2 (mean 2.42) with all but two being a clinical outcome centred publication but with only one paper [18] reporting clinical post-operative outcomes.

### 3.1. Risk of Bias

Risk of bias assessment was undertaken for all included articles using the ROBIN-I tool (Figure 2 and Figure 3). Four studies were excluded from analysis based on these criteria secondary to an identified risk of severe bias impacting outcome assessment [24,25,26,27]. The potential sources of bias in these papers included lack of information of sample population selection [24,25,26,27], missing data with risk of selective reporting or significant confounding of outcome [24,25,26], lack of standardisation of processing between samples [25] and exposure of samples to non-interventional fluids with the introduction of confounding bias [24,25,27].

Within the included works, this tool identified non-critical limitations with confounding [11,12,14,15,20,21], missing data [15], outcome measurement [12,14,15,17] and result reporting [13,18,19]. Four papers were at moderate risk of bias from confounding due to lack of information on the baseline clinical patient status [12,15,21] or unequal number of vein samples taken from patient [11]. Four papers were at moderate risk of bias from reporting of subgroups of the same sample [21] and multiple outcome measures within the outcome domain [13,18,20]. The subjective assessment of outcomes could have been influenced by knowledge of intervention received in four non-blinded studies [12,14,15,17]. All other domains were of low risk of bias.

### 3.2. Endothelial Preservation

Gundry et al. [14], the oldest study reported in this review, established evidence of endothelial damage from analysing EC morphology using scanning electron microscopy (SEM). The group found that normal saline and warm solutions were associated with endothelial damage, independent of each other, but most ECs were well preserved using AWB and this preservation was maximised at cold temperatures (4 degrees Celsius). Direct trauma to the intima or high distention pressure over 100 mmHg was strongly associated with EC injury.

Pimentel et al. [19] also assessed the impact of incubation within AWB and varying distention pressures upon endothelial integrity using SEM. Similar to Gundry et al., they reported that AWB reduced the overall structural damage of endothelium, and that structural damage increased with increasing distention pressure irrespective of incubation solution. These studies, however, may be subject to interpretation bias as they are non-blinded and EC integrity was quantified using SEM which is a subjective visual assessment to quantify integrity loss. They did not explore additional complementary techniques such as expression of endothelial markers or endothelial function to confirm their findings.

Hickethier et al. [15] in a study supporting the use of a buffered culture medium compared to normal saline, utilised SEM, transmission electron microscopy (TEM) and immunohistochemistry (IHC) to establish endothelial integrity after incubation with control and intervention solution of culture medium with albumin (M199 + 20 mmol HEPES). The group identified significant damage to the entire endothelial monolayer with normal saline compared to their intervention (55% loss of ECs vs. 26%, *p* < 0.01) and noted that even areas that appeared to remain intact after incubation were shown to be damaged following TEM assessment. This was not true for the intervention solution which did not demonstrate any morphological changes. These findings were further supported by IHC staining for endothelial marker CD34.

Kurusz et al. [17] compared both AWB and a customised cardioplegia solution ((25 mEq of potassium per litre, pH 7.8, 300 mOsm) with heparinised saline (with solutions kept at 10 degrees Celsius) as well as the role of distension pressure. In line with reports from Dumanski [12], Gundry [14] and Pimentel [19], the authors recognised that veins distended without pressure control showed extensive intimal disruption independent of preservation solution. They also confirmed previous findings regarding improved vein wall integrity with cold solutions [14]. However, in direct conflict with Gundry’s findings [14] that normal saline resulted in wall oedema and endothelial disruption, they reported no major qualitative or quantitative endothelial differences between AWB, NS and cardioplegia samples other than pronounced platelet and white blood cell adhesion where desquamation of normal endothelium had exposed subintimal connective tissue in the AWB group. Understanding that the harvested vein graft experiences significant ischaemia, Kurusz hypothesised that cardioplegia may attenuate endothelial damage through reduction of local metabolic demand and it is a common current-day practice to instil antegrade cardioplegia via vein grafts if maintenance cardioplegia is required during CABG. Whilst no deleterious effects of cardioplegia were identified, no significant benefit was derived either. High osmolarity and non-buffered pH solutions have known associations with endothelial damage [28], which may counteract any metabolic benefit gained from some cardioplegic solutions and could explain why benefit was not seen in this study.

Toto et al. [21] looked at the impact of incubation with heparinised AWB versus normal saline control, with an additional sample group incubated within DuraGraft solution. Two subgroups from each solution were incubated for either two or four hours. The study demonstrated reduced apoptosis (measured by immunofluorescence) at 2 h with DuraGraft compared to both AWB and normal saline (Intensity nuclei/intensity area % DuraGraft 10.11 ± 5.81, Heparinized AWB 13.12 ± 7.10 (*p* = 0.193) and NS 19.44 ± 10.68 (*p* = 0.002). Furthermore, the group noted that none of the solutions maintained structural integrity with prolonged incubation time of 4 h (Intensity nuclei/intensity area % DuraGraft 14.98 ± 5.58, Heparinized AWB 16.01 ± 7.23 (*p* = 0.786), normal saline 20.83 ± 10.34 (*p*-0.110)). It is essential to add that the time points used in this study do not reflect the reality of the time veins are kept in solution as during most CABG procedures the veins are implanted much before 4 **h** post-harvest, and most of the cases before 2 h as well.

Although ex vivo analyses have identified that DuraGraft appears to offer protection of EC function from oxidative stress [20,29], its’ role in preserving endothelial integrity is not clear. Perrault et al. [18] attempted to assess the impact of DuraGraft on vein wall thickness (measured by staged CT assessment at one, three and 12 months after CABG) as a proxy of endothelial integrity. At 12 months, DuraGraft-treated SVGs had a significantly smaller mean wall thickness versus their saline-treated counterparts (0.12 mm ± 0.06 versus 0.02 ± 0.31, *p* = 0.02) and a reduced total vessel diameter at the same time point. Clinical outcomes suggested a slight reduction in significant cardiac events post-operatively in the intervention group, but the populations were too small to make an accurate determination of the extent of this reduction. Luminal diameter, wall thickness and vessel diameter are known to influence vein graft patency and flow after CABG. However, considering the complex pathophysiology of IH, it cannot objectively be used as an indicator of EC integrity, especially considering the limitation in terms of spatial resolution and difficulty imaging distal graft portions [30,31]. Therefore, the accuracy of the measurements from which the authors draw their conclusions is uncertain.

Evans et al. [13] compared the percentage increase in EC apoptosis of vein grafts as a marker of endothelial integrity to establish utility of EuroCollins, University of Wisconsin (UW) and Pyruvate (PYR) solutions compared to normal saline and normalised to ‘control’ samples of vein which had been preserved and assessed immediately following harvest. Using TUNEL assay, they calculated percentage of apoptotic cells and note that the worst endothelial preservation appears to be from immersion in normal saline and EuroCollins (% apoptotic cells control 4.03 ± 0.40, NS 5.01 ± 0.44, EuroCollins 5.40 ± 0.75, UW 4.40 ± 0.50, PYR 4.00 ± 0.47 respectively). Both NS and EuroCollins groups demonstrated a 38 ± 3.6% increase in apoptotic cells compared to control (*p* < 0.05). Similarly, UW preserved segments demonstrated a 25 ± 4.8% increase in apoptosis, but the standout group was samples exposed to pyruvate solution; showing a non-statistically significant increase of apoptosis of only 4 ± 3.1% normalised to control sample. This study stored intervention samples in relative hypothermia at 20 degrees Celsius for an hour to emulate the operating theatre environment. Cold storage has previously been demonstrated to enhance endothelial damage [32], although this would not account for variations in extent of apoptosis in this study as the intervention groups were standardised in this regard. Additionally, their findings were in direct contrast to Gundry [14] who reported that EC integrity was maximally preserved utilising cold preservation solutions.

Kocalik et al. [16] was one of the three studies with a shared methodology of outcome assessment. Following either 4- or 24-h incubations in normal saline, plus 10 g/L concentration papaverine for the intervention group, venous rings were exposed to acetylcholine to determine their endothelial-dependent vasorelaxation capacity following a challenge with potassium chloride solution to induce vasoconstriction. The dilatory response was significantly reduced in the control group at both 4 and 24 h (25% versus 81% *p* = 0.001, and 11% versus 75% *p* = 0.001 respectively) although these variations in function were not significant compared to their ‘within group’ baseline (ie., the same group prior to incubation and testing). However, there was major difference in EC viability between groups with those incubated in papaverine showing 92 ± 3.7% and 87 ± 5% viability at 4- and 24-h incubation (*p* = 0.08) compared to only 34 ± 6.5% (*p* < 0.001) and 22 ± 4.5% (p0.03) in the control groups. The use of papaverine in this study appears to improve EC viability and preserve EC dilatory functions better than normal saline, although exposure times in this study are protracted compared to usual clinical exposure times that the vein graft would be subject to during CABG.

Wilbring et al., the only group to evaluate novel storage solution TiProtec, assessed endothelial function through direct measure of vein wall tension with myography across two published studies [22,23]. TiProtec is based upon histidine-tryptophan-ketoglutarate (HTK) solutions commonly used for graft preservation but with additional oxidative buffers, and metabolic intermediates such as aspartate [33]. Their studies identified better preservation of wall tension and endothelial-dependent vasodilation with TiProtec compared to control solution at 90 min, 24 and 96 h. Functional integrity of endothelium is required to maintain endothelium dependent relaxation, and in turn plays a key role in maintaining the vascular reactivity of the vein graft. These findings indicated reduced impairment of the endothelial layer following short or long-term immersion in TiProtec compared to saline.

### 3.3. Endothelial Inflammation and Oxidative Response

Dumanski et al. [12] assessed the contributions of distention pressure to reductions of endothelial integrity. The authors flushed the vein segments with either AWB or normal saline, at a controlled pressure of 300 mmHg, and noted similar damage to the endothelial surface and equal expressions of CD31, VCAM1, P-selectin and ICAM-1. After flushing, veins were then incubated with the same solution they were flushed with. Adhesion molecule expression did not differ significantly between groups but were very significantly greater than a control group that were exposed to neither solution nor pressure. Mean percentage surface expression for control, AWB and normal saline groups were: CD31 70.94%, 62.29% and 59.69% *p* = 0.0002, VCAM-1 11.51%, 40.23% and 42.71% *p* = 0.0000), ICAM-1 12.60%, 48.42% and 50.63% *p* = 0.0258) and P-selectin 12.53%, 67.81% and 71.25% *p* = 0.024. From this, they concluded that at least a proportion of these findings have come from pressure related effects, although potential preservative benefit of AWB compared to normal saline. Notably there were no details provided regarding the harvest technique used and all vein samples were flushed with NS at the beginning of the experiment protocol, which introduces moderate risk of confounding to their findings.

Tekin et al. [20] used DuraGraft as an intervention solution compared to AWB and focused upon an oxidative stress index as an ex vivo proxy for endothelial function. This is representative of the ability of the vein graft to resist ischaemia-reperfusion injury, and its ability to neutralise radical oxygen species. The authors demonstrated a lower oxidative stress index (OSI) and improved antioxidant ability with DuraGraft, compared to AWB (*p* < 0.0001) suggesting that DuraGraft usage can facilitates a higher capacity for combating ischaemic oxidative stress and would in theory provide better protection against EC dysfunction.

Chen et al. [11] compared six groups of vein segments; three groups each exposed to either normal saline or AWB solution for three different durations; Groups I, II and III. On immunohistochemistry staining, they demonstrated that CD31 and eNOS staining did not differ between groups (Group I (30 ± 15 min incubation) *p* = 0.902, 0.834, Group II (90 ± 15 min incubation) *p* = 0.754, 0.602, Group III (150 ± 15 min incubation) *p* = 0.327, 0.327 respectively) suggesting that neither NS or AWB significantly damaged ECs. Although further assessment of eNOS expression with Western blotting did show stronger expression within the AWB group (Group I: *p* = 0.046; Group II: *p* = 0.006; Group III: *p* = 0.012). eNOS is a key locally produced source of vaso-activity, essential for vascular tone and along with NO, has been shown in mouse models to mitigate key components of VGD including IH and atherosclerosis [3]. Enhanced expression in this group may therefore represent better preservation of key functions of the vein that reduce the risk of later VGD development.

When investigating hypoxic responses that may be triggered by low micro-environmental oxygen, the group noted on Western blotting that the incubation in AWB, and with prolonged incubation times, resulted in a higher expression of the hypoxic marker, hypoxia inducible factor 1. This suggests that AWB, independent of incubation time or time since vein harvest, likely promotes a hypoxic microenvironment, with the known implications of oxidative stress upon vein grafts [3].

Furthermore, the group looked at NF-kB, as a mediator of inflammation and noted a significant increase following incubation in AWB (Group I: *p* = 0.001; Group II: *p* = 0.005; Group III: *p* = 0.009). Whilst NF-kB has been implicated in VGD onset and its expression is commonly seen in response to shear stress. NF-kB upregulation has also been demonstrated as an EC oxidative stress response resulting in pro-inflammatory activation and monocyte recruitment [1]. Unlike others discussed here, this study did not expose ex vivo vein segments to pressure and so their findings demonstrate upregulation via none flow-mediated pathways; more so in the AWB group. Moreover, the authors identified upregulation of glutathione peroxidase (GPx), a key antioxidant enzyme, in the AWB groups (Group I: *p* = 0.294; Group II: *p* = 0.027; Group III: *p* = 0.021) which appears to constitute an attempt of these venous samples to mount an antioxidant defence. This study concluded that AWB may not provide protection from ischaemia-reperfusion type injuries, compared to their NS control samples. However, upregulation of these proteins (NF-kB and GPx) which have been implicated in an EC antioxidant response, may represent an enhanced antioxidant response to a more hypoxic environment when submerged in AWB compared to NS, or may represent better preservation of EC function, and therefore EC mediated antioxidant response, compared to NS exposure.

### 3.4. Survey of National Practice

There were 120 responses received from 100% of UK NHS/HSC adult cardiac surgery units (***n*** = 35, median responses per unit 2). There were eight commonly used preservation fluids used during and after vein harvest *(*Figure 4) comprising whole blood, saline, papaverine and DuraGraft in varying combinations. Most units performed both conventional and endoscopic vein harvest in a surgeon-dependent fashion, except for six units who do not utilise endoscopic vein harvest.

When asked the impact of preservation fluid and additive choice (Figure 5), 21.8% (*n* = 26) said they do not believe this can impact vein graft integrity, and 42% (*n* = 50) do not believe it impacts clinical post-operative outcomes. In addition, 33.6% (*n* = 40) did not believe vein integrity can impact its patency as a conduit. A similar proportion believe it could (43.7%, *n* = 52) and many were not sure (22.7%, *n* = 27). A minority of responders believe that vein integrity can impact short-term clinical outcomes (less than 1 year) (27.7%, *n* = 33) but this proportion increased regarding the vein integrity impacting long-term outcomes beyond a year (35.5%, *n* = 42). Interestingly, an overwhelming proportion of responders (61.3%, *n* = 73 vs. 1.7%, *n* = 2) believe that clinical outcomes are more important than binary graft patency although many report that both factors are equally important (*n* = 42, 35.3%).

Most responders reported differences between surgeons in their unit in their preferred choice of technique for vein harvest and preservation fluid (*n* = 98, 82.4%) (Figure 6). A minority of responders (*n* = 6, 5%, from different centres) are aware of a departmental policy on the use of preservation fluids but most centres do not have an active policy (*n* = 69, 58%) or the surgical team are not aware of one (*n* = 44, 37%). From this we can presume if a policy exists, it is not widely implemented. Interestingly, it seems that many units do not offer objective intra-operative quality assessment of grafts after CABG e.g., intra-operative Doppler ultrasound; only 29 respondents (24.4%) reported it being used in their unit, with reasons for it not being used cited as clinical preference, lack of resources or perceived lack of clinical utility, which is surprising considering the evidence supporting the use of such objective measures [34].

## 4. Discussion

CABG is the standard of care for severe coronary artery disease. However, the long-term success of CABG is limited by vein graft disease comprising accelerated atherosclerosis that leads to vein graft failure and MACE including acute coronary syndromes, the need for repeat revascularisation, and death. MACE occur in up to 30% of CABG patients within 5 years [35,36,37].

Although arterial grafts are known to have better patency than vein grafts [38,39], RCTs have not consistently demonstrated improved long-term outcomes with the use of multiple arterial grafts versus LITA to LAD plus LSV grafts [40,41,42]. LSVs have more reliable handling characteristics and are readily available. As a result, over 90% of the 20,000 CABG procedures performed annually in the UK use the LSV and the LSV remains the most frequently used conduit by volume for CABG [43]. Venous trauma during harvest is thought to contribute chiefly to acute thrombosis, which occurs in up to 5% of human and large animal vein grafts, but not to vein graft atherosclerosis. However, endothelial integrity has a demonstrated critical role in supporting arterialisation of vein grafts; a process required for their adaptation into the coronary circulation and requirement for prolonged patency of venous grafts [44]. Chemical or physical injury to the venous endothelium during harvest or preservation promotes vascular inflammation within the vein wall. In turn this reduces the veins vaso-activity and resistance to oxidative stressors. Experimental vein grafts demonstrate acute inflammatory responses within hours of grafting [44,45]. Levels of cell apoptosis increase by 24 h and then decline, followed by a myeloproliferative response that leads to wall thickening within weeks, due in part to intimal hyperplasia. Progressive remodelling, cellular proliferation, extracellular matrix accumulation, and wall thickening continue for up to 6 months [46].

There have been many studies addressing the use of preservation solutions upon vein graft outcome measures, but gross heterogeneity in type of solutions, study design and outcome measures assessed has led to failure of consensus amongst surgeons. Within this review, most studies have demonstrated that normal saline appears to have negative effects on endothelium, or their critical functions [14,15,16,18,19,20,21,22,26]. Others identified no observable difference between control and intervention solutions but did not find saline to be associated with improved outcome measures [12,17].

Whilst not included in this review, this conclusion has also been mirrored in studies addressing clinical post-operative outcomes such as the large PREVENT IV trial [47] which identified lower rates of vein graft failure with the use of buffered saline compared to normal saline or AWB. This study is however limited by the protocol used in the process of preparing conduits as part of the intervention used.

The most effective preservation solution interventions identified in this review were TiProtec and DuraGraft [18,20,21,22,23]. These solutions, particularly when combined with hypothermia, were shown to preserve endothelial-dependent vaso-activity. The major limitation of validating these findings is the difficulty of reproducibility of study parameters within clinical practice; prolonged immersion of preservation fluid for several hours or at specific below-ambient temperatures is logistically challenging to replicate in the operating room without compromising other aspects of patient care. Only one study specifically addressed changes EC integrity through adhesion molecule expression by solute [12]; these inter-endothelial junctions have a key role in regulating endothelial barrier functions [48]. Moreover, only one study focused observed vein patency using cross-sectional imaging [18]. Computed Tomographic Angiography (CTA) of coronary arteries is an accurate method of assessing graft patency in-situ although an imperfect assessment tool as its resolution cannot assess small degrees of stenosis in patent veins, is useful mainly in proximal graft segments and results in radiation exposure to patient [49]. Using more invasive imaging methods, such as intra-vascular ultra-sound and optical coherence tomography, which are considered accurate and validated invasive methods for the evaluation of vein graft IH [50,51,52] may be of greater value if we wish to accurately assess the degree of IH and changes within the different layers of vein graft wall.

There are other issues to consider when studying the impact of preservation fluids on EC or outcomes such as the temperature of the solution and the harvest techniques. In most studies, the samples were stored at room temperature to emulate that of the operating room but the two studies addressing lower temperatures [14,17] suggest that hypothermic solutions may be beneficial in maintaining endothelial integrity. Harvesting technique is recognised to impact EC integrity and can affect patency and outcomes [53,54,55]. The three techniques reported within included studies included the open, no-touch and endoscopic techniques. The choice of harvest technique to be used is largely determined by the primary surgeon or harvester preference. Overall, the evidence of superiority of the various techniques available is largely equivocal, as findings are variable and dependent upon which outcome parameters are used within specific studies [56,57]. Importantly, most studies identify that experience of the clinician harvesting the vein is one of the most influential factors associated with better or worse preservation of the vein [54]. Crucially therefore, most papers included within this review recognised this and ensured standardisation and adequate experience of the individual harvesting veins for grafting for use in these studies. Finally, no studies utilising preservation solution additives explored dose-dependence relationships. Their published reports did not discuss how their dosing of additives was chosen and variations of these doses may change the impact on the venous endothelium; it may well be that additives such as papaverine or pyruvate could be useful but the current dosing regimens tested are suboptimal.

In the absence of strong recommendations within the literature, we sought to explore the scope of practices in the UK and the understanding of the role of preserving EC integrity on outcomes and patency, we wanted to explore the views of UK surgeons on what they deemed the most important parameter, patency, or outcomes. Although the heterogeneity reported within our survey was anticipated, it was surprising that a third of responders did not believe that the integrity of the vein and its endothelium could impact conduit patency, particularly in the short-term. Most responses identified clinical outcomes as more important than graft patency in isolation which on the surface can be correct, but it ignores the fact that patency is a key driver of outcomes when grafting large and important coronary vessels.

### Limitations

In this study there was significant heterogeneity between studies due to the use of different outcome assessment measures and varying methods of conducting these measurements. This heterogeneity and resultant lack of consistency in reported outcomes forwent meta-analysis. Additionally, the limited availability of robust randomised data as well as limited clinical validation of findings limits the generalisability of the conclusions of these studies upon the broader population of patients undergoing LSV harvest for CABG.

## 5. Conclusions

There is substantial heterogeneity in both practice across the UK and reporting within the literature. Buffered preservation solutions, particularly novel ‘for-purpose’ solutions DuraGraft and TiProtec, are probably superior at preserving EC integrity and function, compared to commonly used non-crystalloid preservation solutions such as AWB, and they are superior to saline alone. Current published evidence does not facilitate generation of robust data to support changes to clinical practice at present and there is a need for prospective interventional studies to address this common feature of practice to facilitate improvement of long-term patency of venous conduits. The detrimental impact of some vein preservation solutions on graft integrity and clinical outcomes is recognised but not fully characterised, and so should also be addressed through these robust prospective clinical trials.

## Figures and Tables

**Figure 1 cells-12-00815-f001:**
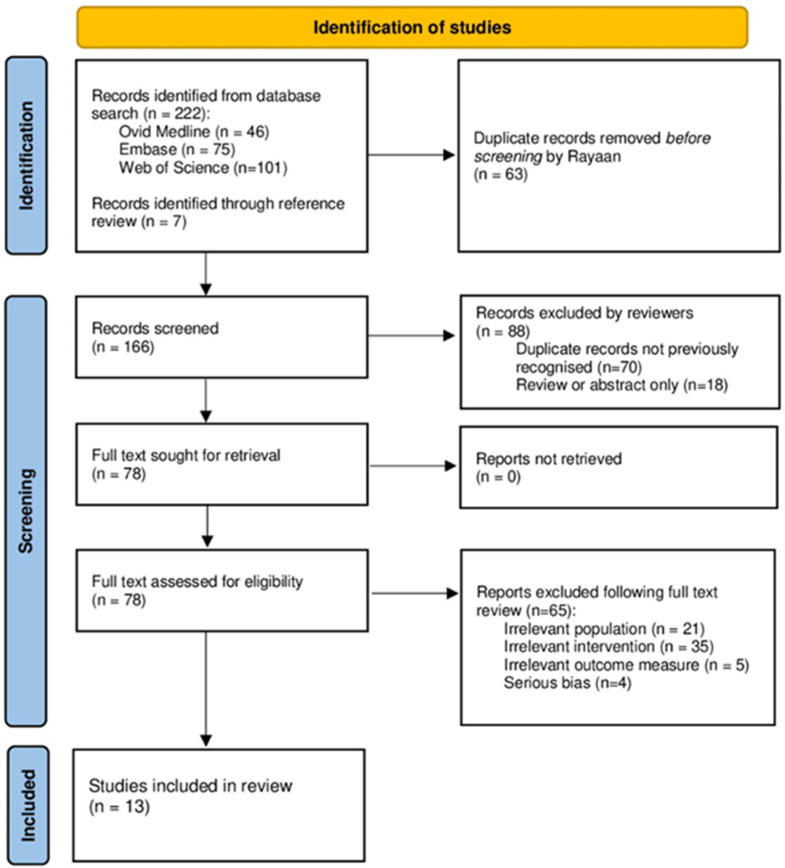
PRISMA flowchart detailing the summary of systematic literature search, screening and included paper selection (*n* = number of studies).

**Figure 2 cells-12-00815-f002:**
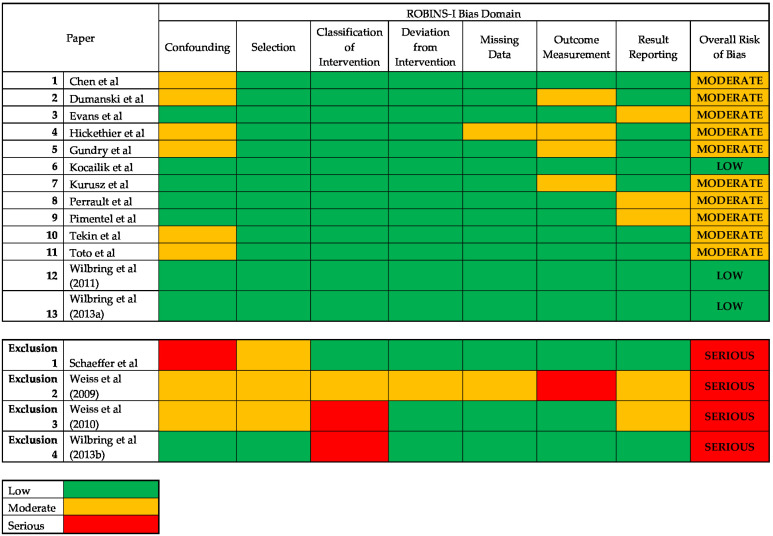
ROBINS-I tool assessment for risk of bias for all studies meeting inclusion criteria [11,12,13,14,15,16,17,18,18,19,21,22,23,25,26,27].

**Figure 3 cells-12-00815-f003:**
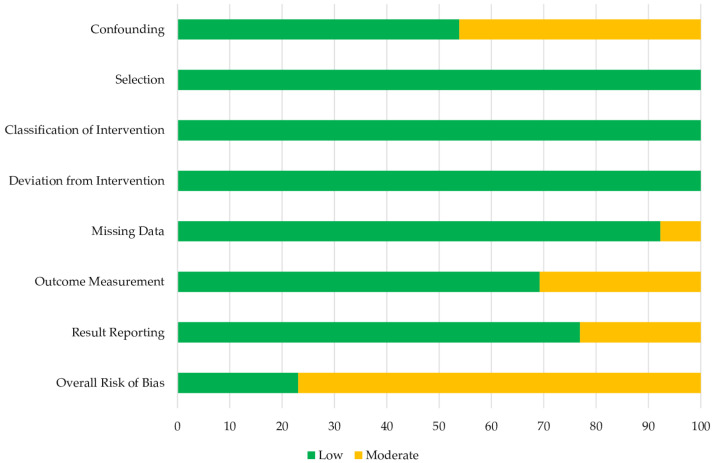
Summary plot of risk of bias for included studies.

**Figure 4 cells-12-00815-f004:**
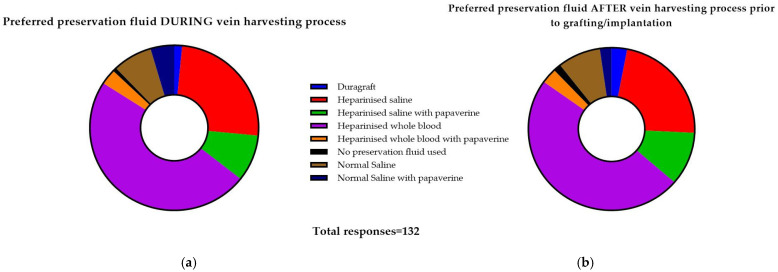
(**a**)—(left) and (**b**)—(right). Choice of preservation fluid during vein graft harvest (**a**) and after harvest, prior to implantation (**b**) (total responses; some respondents reported using more than one fluid in their unit/practice.

**Figure 5 cells-12-00815-f005:**
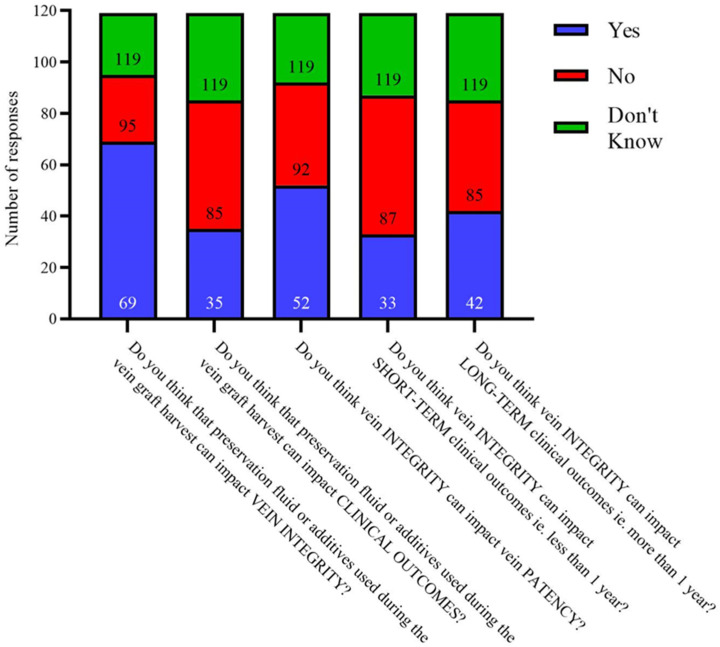
Respondents answers when asked about the impact of preservation fluid upon outcomes and graft integrity.

**Figure 6 cells-12-00815-f006:**
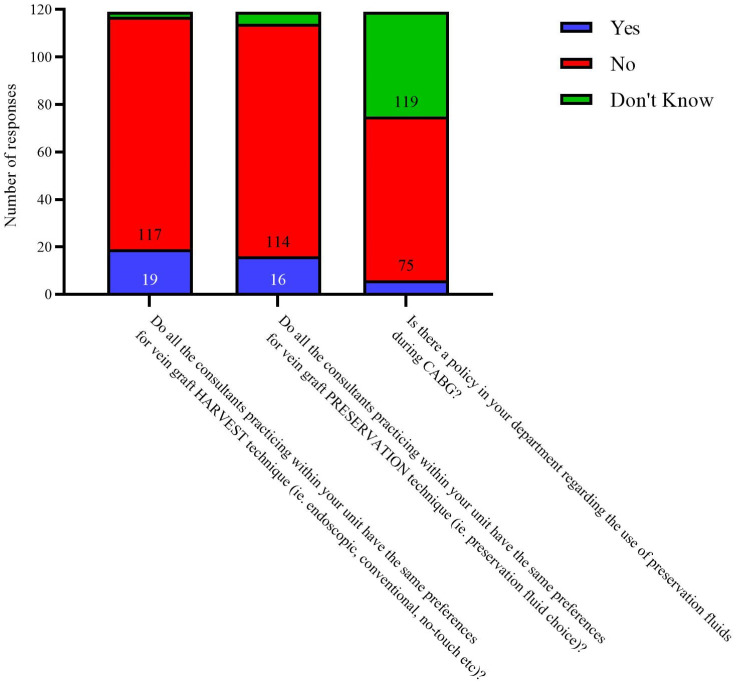
Reported consultant preferences within units.

**Table 1 cells-12-00815-t001:** Characterisation of included studies.

Reference	Study Design	Patient Population	Total Number of Vein Samples Studied	LSV Harvest Method	Control Solution	Intervention Solution(s)	Additional Intervention(s)	Assessing	Primary Outcome(s)	Secondary Outcome
Chen et al. [11]	Single centre, randomised cohort study	21	162	Open and endoscopic	Normal saline	Heparinized whole blood	Time of solution exposure	Endothelial integrity	Staining of endothelial dependent markers	Oxidative stress
Dumanski et al. [12]	Single centre, prospective cohort study	48	144	No information provided.	Normal saline	Heparinized whole blood	Distention pressure	Endothlial integrity	Staining of endothelial dependent markers	Nil
Evans et al. [13]	Single centre, prospective cohort study	14	70	Open	Normal saline	Eurocollins, University of Wisconsin, Pyruvate	Nil	Endothelial integrity	Endothelial apoptosis	Nil
Gundry et al. [14]	Single centre, prospective cohort study	30	35	“Gentle dissection”	Heparinized Normal saline	Heparinized whole blood	Temperature and distention pressure	Endothelial integrity	Composite score of endothelial morphology	Nil
Hickethier et al. [15]	Single centre, prospective cohort study	6	Unknown	Open, no-touch	Normal saline	Buffered cell culture solution with albumin	Nil	Endothelial integrity	Endothelail monolayer integrity	Staining of endothelial dependent markers
Kocailik et al. [16]	Single centre, randomised cohort study	80	80	Open, no-touch	Normal saline	Papaverine	Time of solution exposure	Endothelial function	Endothelial dependent wall relaxation	Nil
Kurusz et al. [17]	Single centre, prospective cohort study	5	25	No information provided.	Normal saline	Heparinized whole blood, Custom cardioplegia solution	Temperature and distention pressure	Endothelial integrity	Endothelail monolayer integrity	Nil
Perrault et al. [18]	Multi centre, blinded, randomised control trial	125	250	Open and endoscopic	Heparinized Normal saline	Duragraft	Proximal vs. distal position in vessel	Endothelial integrity	Wall thickness	Nil
Pimentel et al. [19]	Single centre, prospective cohort study	12	42	Open	Normal saline	Heparinized whole blood	Distention Pressure	Endothelial integrity	Composite score of endothelial morphology	Nil
Tekin et al. [20]	Single centre, prospective cohort study	50	150	“Standard hospital protocol”	Normal saline	Heparinized whole blood, Duragraft	Nil	Endothelial integrity	Oxidative stress index and total antioxidant status	Nil
Toto et al. [21]	Single centre, prospective cohort study	12	72	Open	Normal saline	Heparinized whole blood, Duragraft	Time of solution exposure	Endothelial integrity	Endothelial cell apoptosis	Nil
Wilbring et al. (2011) [22]	Single centre, prospective cohort study	Unknown	19	Open	Normal saline	TiProtec	Nil	Endothelial function	Endothelial dependent wall relaxation	Vasodilation
Wilbring et al. (2013) [23]	Single centre, prospective cohort study	19	38	Open	Normal saline	TiProtec	Time of solution exposure	Endothelial function	Endothelial dependent wall relaxation	Nil

**Table 2 cells-12-00815-t002:** Composition of preservation solutions reported within included studies.

Solution	Components	Additive Concentration(s) [mmol/L] if Applicable	Osmolality
Normal saline	Salt solution	154 sodium chloride	308 [mosmol/L]
Heparinized whole blood	Autologous whole blood with heparin sodium	Variable heparin sodium dosage dependent on local preferences	289 ± 3–302 ± 5 [mmol/kg]
DuraGraft (Somahlution Inc., Jupiter, FL, USA)	Physiological salt solution with additives	Not reported publically.	Not reported publically.
TiProtec (Kohler Chemie, Germany)	Salt solution with additives	14 sodium chloride, 73 potassium chloride, 8 magnesium chloride, 1 Disodium hydrogen phosphate, 0.05 Calcium chloride • 2 H_2_O, 30 N-acetyl histidine • H_2_O, 2 Tryptophan, 2 α-Ketuglutaric acid, 5 Asparagine acid, 10 Glycine, 5 Alanine, 20 Sucrose, 10 Glucose monohydrate, 0.1 Deferoxamine mesylate, 0.02 3,4-Dimethoxy-N-methylbenzohydroxamic acid	307 [mosmol/L]
EuroCollins (Baxter Healthcare, Old Toongabbie NSW, Australia)	Physiological salt solution with additives	0.11 potassium dihydrogen phosphate, 0.54 Dipotassium hydrogen phosphate trihydrate, 0.06 potassium chloride, 0.05 sodium bicarbonate, 0.006 procaine hydrochloride, 195 glucose,	375 [mosmol/L]
University of Wisconsin (Belzer UW, Bridge to Life, Northbrook, IL, USA)	Salt solution with additives	Pentafraction, 1.99 Lactobionic Acid (as Lactone), 0.19 Potassium Phosphate monobasic, 0.07 Magnesium Sulfate heptahydrate, 0.97 Raffinose pentahydrate, 0.07 Adenosine, 0.01 Allopurinol, 0.05 Total Glutathione, 0.31 Potassium Hydroxide, Sodium Hydroxide/Hydrochloric Acid to adjust to pH 7.4	320 [mosmol/L]
Pyruvate	Supplement for cell culture medium	100 Sodium Pyruvate	165–205 [mosmol/L]
Medium 199 with HEPES (Thermo Fisher Scientific, Inc., Waltham, MA, USA)	Buffered cell culture medium	0.67 Glycine, 0.28 L-Alanine, 0.33 L-Arginine hydrochloride, 0.23 L-Aspartic acid, 5.68 L-Cysteine hydrochloride-H_2_O, 0.11 L-Cystine 2HCl, 0.51 L-Glutamic Acid, 0.68 L-Glutamine, 0.10 L-Histidine hydrochloride-H_2_O, 0.08 L-Hydroxyproline, 0.31 L-Isoleucine, 0.46 L-Leucine, 0.38 L-Lysine hydrochloride, 0.10 L-Methionine, 0.15 L-Phenylalanine, 0.35 L-Proline, 0.24 L-Serine, 0.25 L-Threonine, 0.05 L-Tryptophan, 0.22 L-Tyrosine disodium salt dihydrate, 0.21 L-Valine, 2.84 × 10^−4^ Ascorbic Acid, 4.09 × 10^−5^ Biotin, 0.003 Choline chloride, 2.1 × 10^−5^ D-Calcium pantothenate, 2.27 Folic Acid, 5.81 Menadione (Vitamin K3), 2.05 Niacinamide, 2.03 Nicotinic acid (Niacin), 3.65 × 10^−4^ Para-Aminobenzoic Acid, 1.23 × 10^−4^ Pyridoxal hydrochloride, 1.21 × 10^−4^ Pyridoxine hydrochloride, 2.66 × 10^−5^ Riboflavin, 2.97 × 10^−5^ Thiamine hydrochloride, 3.05 × 10^−4^ Vitamin A (acetate), 2.52 × 10^−4^ Vitamin D2 (Calciferol), 1.80 alpha Tocopherol phos. Na salt, 2.78 × 10^−4^ i-Inositol, 1.80 Calcium Chloride (CaCl_2_) (anhyd.), 0.001 Ferric nitrate (Fe(NO_3_)-9H_2_O), 0.81 Magnesium Sulfate (MgSO4) (anhyd.), 5.33 Potassium Chloride (KCl), 26.19 Sodium Bicarbonate (NaHCO_3_), 105.17 Sodium Chloride (NaCl), 1.01 Sodium Phosphate monobasic (NaH2PO4) anhydrous, 0.003 2-deoxy-D-ribose, 0.02 Adenine sulfate, 5.76 Adenosine 5′-phosphate, 0.001 Adenosine 5′-triphosphate, 5.17 × 10^−4^ Cholesterol, 5.56 D-Glucose (Dextrose), 1.63 Glutathione (reduced), 0.001 Guanine hydrochloride, 25.04 HEPES, 0.002 Hypoxanthine Na, 0.05 Phenol Red, 0.003 Ribose, 0.61 odium Acetate, 0.002 Thymine, Tween 80^®^, 0.002 Uracil, 0.002 Xanthine-Na	270–330 [mmol/kg]
Customised cardiplegia		140 sodium, 25 potassium, 3 magnesium, 104 chlorine, 27 acetate, 23 gluconate, 1.36 calcium, 23 bicarbonate radical	300 [mosmol/L]
Papaverine	Papaverine in 0.9% sodium chloride	2.95 × 10^−6^ Papaverine, 154 Sodium chloride	Unknown

## Data Availability

Anonymised iterations of the data contained within this article are available upon reasonable request to the corresponding author. They are not publicly available due to the privacy of survey responders.

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
