# Peer review of "The Role of Preservation Solutions upon Saphenous Vein Endothelial Integrity and Function: Systematic Review and UK Practice Survey"

_cells, 2023, doi:10.3390/cells12050815_

Round 1
Reviewer 1 Report
Despite limited data for analysis and essentially indirect support for the conclusions, the relevance of the topic to coronary artery bypass grafting is high, warranting publication of the report. There are some minor grammatical errors that should be apparent to the copy editor.
Author Response
Thank you for your comments.
Reviewer 2 Report
I read with interest the review by Layton et al on the role of preservation solutions upon saphenous vein endothelial integrity and function including a UK practice survey. This is a well-performed review according to the guidelines of PRISMA standard. Despite the significant heterogeneity among studies the conclusions are supported by the analysis of the included studies. The UK practice survey is also useful and confirms the heterogeneity in UK practice as well. The review is well-structured and the figures are informative. I found only one minor missing data in line 130 : “… between 16th – 16th September 2022 …”, where it seems that is missing the month of the starting date.
Author Response
Thank you for your comments. Thank you for highlighting the error in line 130; this has now been amended.
Reviewer 3 Report
In the current manuscript Layton et al. give an excellent overview on the role of preservation solutions upon saphenous vein endothelial integrity and function during vein graft surgery. The extensive literature as well as the UK broad survey is impressive, showing the heterogeneity in choice and use of preservation fluids. The paper is well written and structured, but the authors might take a helicopter view and not just give a factual sum-up but als provide some conclusion on best practices which would be helpful for the readers.
Major concerns
The authors may be a bit more informative in the abstract by indicating what is the most frequently used preservation solution, or which seems to give the best results. The current concluding sentences of the abstract - ……the quality of existing evidence was low. There is an unmet need for high quality trials evaluating the potential for these interventions to improve long-term patency in venous bypass grafts. – is very cautious and not really stimulating to read to rest of the paper. Please be more informative and precise. Consider tocopy lines 452-453 into the abstract.
For instance , indicate in the abstract that for the survey it appears that about 50% of the respondents in the UK currently uses Duragraft as preservation fluid.
Including a table of the main preservation fluids used in the main body of the manuscript would strengthen this manuscript. This is too important for only including it in a supplement
Lines 165-171 :
Table 1 and 2 are lacking in the version of the manuscript I got to review. Please include them in the main body.
Minor :
Line 130 : correct statement : “ between 16th – 16th September 2022”,
Author Response
Thank you for your comments.
Regarding our summary of findings, each trial showed great heterogeneity which I believe forgoes any strongly held conclusion regarding a specific solution as the findings are largely dependent on the study conditions. However, we agree that the abstract should be amended to be more informative to the reader and encourage them to read the full manuscript. I have therefore made edits in-line with your suggestions.
The revised abstract now reads (from line 21) “Most studies demonstrated that normal saline appears to have negative effects on venous endothelium and the most effective preservation solutions identified in this review were TiProtec and DuraGraft. The most used preservation solutions in the UK are heparinized saline or autologous whole blood. There is substantial heterogeneity in practice and reporting of trials evaluating vein graft preservation solutions, and the quality of existing evidence was low. There is an unmet need for high quality trials evaluating the potential for these interventions to improve long-term patency in venous bypass grafts.”
Lines 165-171 : We agree that this should be included in the main body of the paper. Unfortunately, the journal submission and formatting requirements precluded this being included in the main manuscript file and it was uploaded as supplementary materials. However, it will be within the body of the main text in the published works.
Line 130 : thank you for highlighting this error. It has now been amended.